# Comprehensive analyses reveal the role of histone deacetylase genes in prognosis and immune response in low-grade glioma

Lin Shen[1], Yanyan Li[2], Na Li[1], Liangfang Shen[1], Zhanzhan Li[1,3]*

1 Department of Oncology, Xiangya Hospital, Central South University, Changsha, Hunan Province, P.R. China, 2 Department of Nursing, Xiangya Hospital, Central South University, Changsha, Hunan Province, P. R. China, 3 National Clinical Research Center for Geriatric Disorders, Xiangya Hospital, Central South University, Changsha, Hunan Province, P.R. China

* lizhanzhan@csu.edu.cn

**Data Availability Statement:** The TCGA data can be obtained from the https://portal.gdc.cancer.gov/repository (LGG mRNA expression profiling), and CGGA data can be available from the Chinese

## Abstract

Many studies have shown that Histone deacetylases (HDAC) is involved in the occurrence of malignant tumors and regulates the occurrence, proliferation, invasion, and migration of malignant tumors through a variety of signaling pathways. In the present, we explored the role of Histone deacetylases genes in prognosis and immune response in low-grade glioma. Using consensus clustering, we built the new molecular clusters. Using HDAC genes, we constructed and validated the prognostic model in two independent cohort datasets. Patients were divided into high-risk and low-risk groups. Then, we explored the molecular characteristics, clinical characteristics, tumor microenvironment and immune infiltration levels of two clusters and risk groups. Receiver operating characteristic analyses were built for model assessment. We finally detected the expression levels of signature genes between tumor and normal tissues. Low-grade can be separated into two molecular clusters using 11 HDACs genes. Two clusters had different clinical characteristics and prognosis. Nex, we constructed a prognosis model using six HDAC genes (HDAC1, HDAC4, HDAC5, HDAC7, HDAC9, and HDAC10), which was also validated in an independent cohort dataset. Furthermore, multivariate cox regression indicated that the calculated risk score was independently associated with prognosis in low-grade glioma, and risk score can predict the five-year survival probability of low-grade glioma well. High-risk patients can be attributed to multiple complex function and molecular signaling pathways, and the genes alterations of high- and low-risk patients were significantly different. We also found that different survival outcomes of high- and low- risk patients could be involved in the differences of immune filtration level and tumor microenvironment. Subsequently, using signature genes, we identified several small molecular compounds that could be useful for low-grade glioma patients' treatment. Finally, we detected the expression levels of signature genes in tumor tissues. our study uncovers the biology function role of HDAC genes in low-grade glioma. We identified new molecular subtypes and established a prognostic model based on six HDAC genes, which was well applied in two independent cohort data. The regulation of HDAC genes in low-grade glioma involved in multiple molecular function and signaling pathways and immune

Glioma Genome Atlas (http://www.cgga.org.cn/: mRNAseq_693 and mRNAseq_325). The expression levels of HDAC genes from non-tumor and glioma patients can be available from the GEO (GSE4290). These data can be available by anyone without any restriction.

**Funding:** This study was supported by the National Natural Science Foundation of China (LZZ: 82003239), and Science Foundation of Xiangya Hospital for Young Scholar (LZZ: 2018Q012), and National Natural Science Foundation of China (SLF:81974466). The funders had no role in study design, data collection and analysis, decision to publish, or preparation of the manuscript.

**Competing interests:** The authors have declared that no competing interests exist.

infiltration levels. Further experiments in vivo and vitro were required to confirm the present findings.

## Introduction

Glioma is the most common brain tumor, accounting for about 25% of primary brain tumors and 80% of primary intracranial malignant tumors [1]. Different grades of gliomas have obvious clinical and histological heterogeneity. Glioma growth is aggressive, and low-grade gliomas often progress to high-grade gliomas and eventually lead to death [2]. According to the classification criteria of the World Health Organization, gliomas can be divided into low-grade gliomas and glioblastomas [3]. Glioblastomas account for 60% of all primary tumors in adults, and the prognosis for glioblastomas is currently poor [4]. Low-grade gliomas include WHO grades I and II, which grow slowly, with insignificant initial symptoms, and are not easily detected clinically, so they are easily overlooked. WHO further classifies low-grade gliomas into IDH-mutated diffuse astrocytoma, IDH wild-type diffuse astrocytoma, and IDH-mutated oligodendrogliomas with 1p19q co-deletion [5]. In the past few decades, although many studies have been carried out on glioma, the current treatment of glioma is still mainly limited to surgery, chemotherapy and radiotherapy, and the prognosis of patients is still unsatisfactory [6]. In clinical practice, decision makers still urgently need to find more effective molecular markers for distinguishing glioma subtypes and assessing prognosis to carry out more precise and individualized treatment.

Histone deacetylases (HDACs) are a class of proteases that play an important role in the structural modification of chromosomes and the regulation of gene expression [7]. In general, the acetylation of histones is conducive to the dissociation of DNA and histone octamers, and the relaxation of nucleosome structure, so that various transcription factors and co-transcription factors can specifically bind to DNA binding sites and activate genes transcription. In the nucleus, histone acetylation and histone deacetylation are in dynamic equilibrium, and are jointly regulated by histone acetyltransferase (HAT) and histone deacetylase (HDAC) [8]. HAT transfers the acetyl group of acetyl-CoA to specific lysine residues at the amino terminus of histones, HDAC deacetylates histones, binds tightly to negatively charged DNA, chromatin is dense and coiled, and gene transcription is inhibited [9]. In cancer cells, the overexpression of HDACs leads to enhanced deacetylation, which increases the attraction between DNA and histones by restoring the positive charge of histones, making the relaxed nucleosomes very tight, unfavorable for specific genes expression, including some tumor suppressor genes [10, 11]. Many studies have found that abnormal expression of HDAC is related to the occurrence and development of tumors. HDAC includes four classes: class I (HDAC1, HDAC2, HDAC3, HDAC8), class II (HDAC4, HDAC5, HDAC6, HDAC7, HDAC9, HDAC10), class III (SIRT1-SIRT7), and class IV (HDAC11) [7]. Various inhibitors have been developed for HDAC. Histone deacetylase inhibitors (HDACi) can regulate the expression of apoptosis and differentiation-related proteins by increasing histone acetylation in specific regions of chromatin and stability, induce apoptosis and differentiation, and become a new class of antitumor drugs [12]. Previous studies have explored the role of HDAC genes in some tumors, however the function of HDAC-related genes in low-grade gliomas is unclear [13]. In the current study, we comprehensively analyzed the impact of HDAC-related genes on the prognosis and immune function of low-grade glioma, and our study provides new insights for individualized treatment of low-grade glioma.

## Materials and methods

This study followed the Tripod checklist prediction model development and validation (S1 Table).

### Data acquisition

The present study consisted of training dataset from The Cancer Genome Atlas (TCGA) and validations dataset from the Chinese Glioma Genome Atlas. The expression levels of HDAC genes from non-tumor and glioma patients can be available from the GEO (GSE4290). We obtained these data on June 21, 2021. These data did not include any identified individual information. We also downloaded the gene mutation from TCGA.

We used the genomics of drug sensitivity in cancer to explore the chemotherapy sensitivity. We identified 11 HDAC genes from the molecular signatures database: HDAC1, HDAC2, HDAC2, HDAC4, HDAC5, HDAC6, HDAC7, HDAC8, HDAC9, HDAC10, HDAC11. As the data was downloaded from TCGA, further approval by an ethics committee was not needed. Data processing was performed in accordance with the TCGA human subject protection and data access policies.

### Identification of molecular clusters

We identified the molecular clusters using consensus cluster plus methods, which can include negative value in expression profiling. After 1000-time calculation via extracting 80% of sample size, we finally identified the optimal number of clusters. Principle component analysis (PCA) and t-distributed stochastic neighbor embedding (tSNE) were used for showing distribution of molecular clusters. This process is carried out using "Consensus Cluster Plus" R package [14, 15].

### Construction and validation of the prognostic model based on HDAC genes

The primary outcome was overall survival (OS). We identified the signature genes using the Least Absolute Shrinkage and Selection Operator (LASSO) regression, and the regression coefficients were calculated. Next, we give a risk score for each sample of training dataset (TCGA) and validation dataset (CGGA) using the following formula: $\beta_1{}^*Exp(1)+\ldots+\beta_n{}^*Exp(n)$. We then separated patients into high-risk group and low-risk groups using the median of risk score. Kaplan-Meier analysis was used to compare the survival curve of high- and low-risk groups. The time-independent receive operating characteristics curve (ROC) was used for evaluating predictive ability at one-year, two-year and three-year OS. PCA analysis was used to identify the risk distributions. We validated these analyses in CGGA dataset.

### Clinical correlation and independent analysis

To explore the effect of clinical parameters on model, we performed the stratified analyses in clinical stratifications. We also compared the risk scores of different clinical parameters using "limma" R packages. The univariate and multivariate cox regression was used for identifying risk score was an independent predictor for OS in low-grade glioma. These parameters included age, gender, primary and recurrent grade, history of chemotherapy and radiotherapy, IDH mutant status, and 1p19q co-deletion status. We used the area under the curve (AUC) of ROC to evaluated the predictive ability of risk score and other clinical parameters. Finally, we built the nomogram to estimate the individual's prognosis risk at 1-year, 3-year, and 5-year.

The calibrations plot was used to evaluate fitting degree between predictive probability and actual probability.

## Functional, pathway enrichment and mutations analysis

To explore the molecular characteristics, we performed GO functional enrichment and KEGG pathways analyses using "clusterProfiler" R package. Then, we analyzed the gene alteration frequencies, variation classification, variant type, and co-occurrence and mutually exclusive between high- and low-risk groups.

## Tumor microenvironment, Immune filtration, and drug sensitivity analysis

We compared the ESTIMATE, stromal, immune scores of high- and low-risk groups using ESTIMATE algorithm. Next, we evaluated the immune infiltration levels (immune cell and immune function) in different risk groups. To explore the chemotherapy sensitivity of signature genes, we calculated the Pearson correlation coefficients. $|R|>0.25$ and $P<0.05$ were considered significantly correlated.

## Validation of HDAC genes in glioma and non-tumor tissue

To validate the expression of signature HDAC genes (HDAC1, HDAC4, HDAC5, HDAC7, and HDAC9, HDAC10), we analyse the expression levles of these HDAC genes in low-grade glioma and non-tumor tissues. This mRNA expression data was from GSE4290. The tissue collection was approved by the NCI IRB committee with informed consent obtained from all subjects [16].

## Statistical analysis

Differences for category variables were performed using Chi-square test. The comparisons of OS curve were achieved using log-rank test. T test was used for comparing the differences of HDAC genes expression between normal and tumor glioma. All statical analysis were finished using R software 4.0.2, and $P<0.05$ was considered significant level.

# Results

## Identification of molecular clusters in low-grade glioma

The whole data processing was presented in Fig 1A. We first explored the correlations among these HDAC genes and found HDAC3 was positively associated with HDAC1, HDAC6, HDAC7, HDAC8, and was negatively with HDAC11. HDAC4 showed significantly negative associations with HDAC1 and HDAC7. HDAC5 was positively associated with HDAC2, and HDAC8, and HDAC10 but HDAC7. HDAC11 showed negative associations with other HDAC genes (Fig 1B). Using mRNA expression profiling of 11 HDAC genes, we performed the consensus clusters and identified two molecular clusters (Cluster 1 and Cluster 2, Fig 1C and S2 Table). PCA and tSNE also showed two different components distributions (Fig 1D and 1E). The Kaplan-Meier survival curve indicated that the cluster 2 had worse OS than cluster 1 (Fig 1F). Two clusters showed different clinical characteristics in grade and survival outcomes (Fig 1G). Heatmap indicated that HDAC1, HDAC3, HDAC6, HDAC2, HDAC8 and HDAC10 were highly expressed in Cluster 2 while HDAC11, HDAC3, HDAC4, wand HDAC5 were highly expressed in Cluster 1.

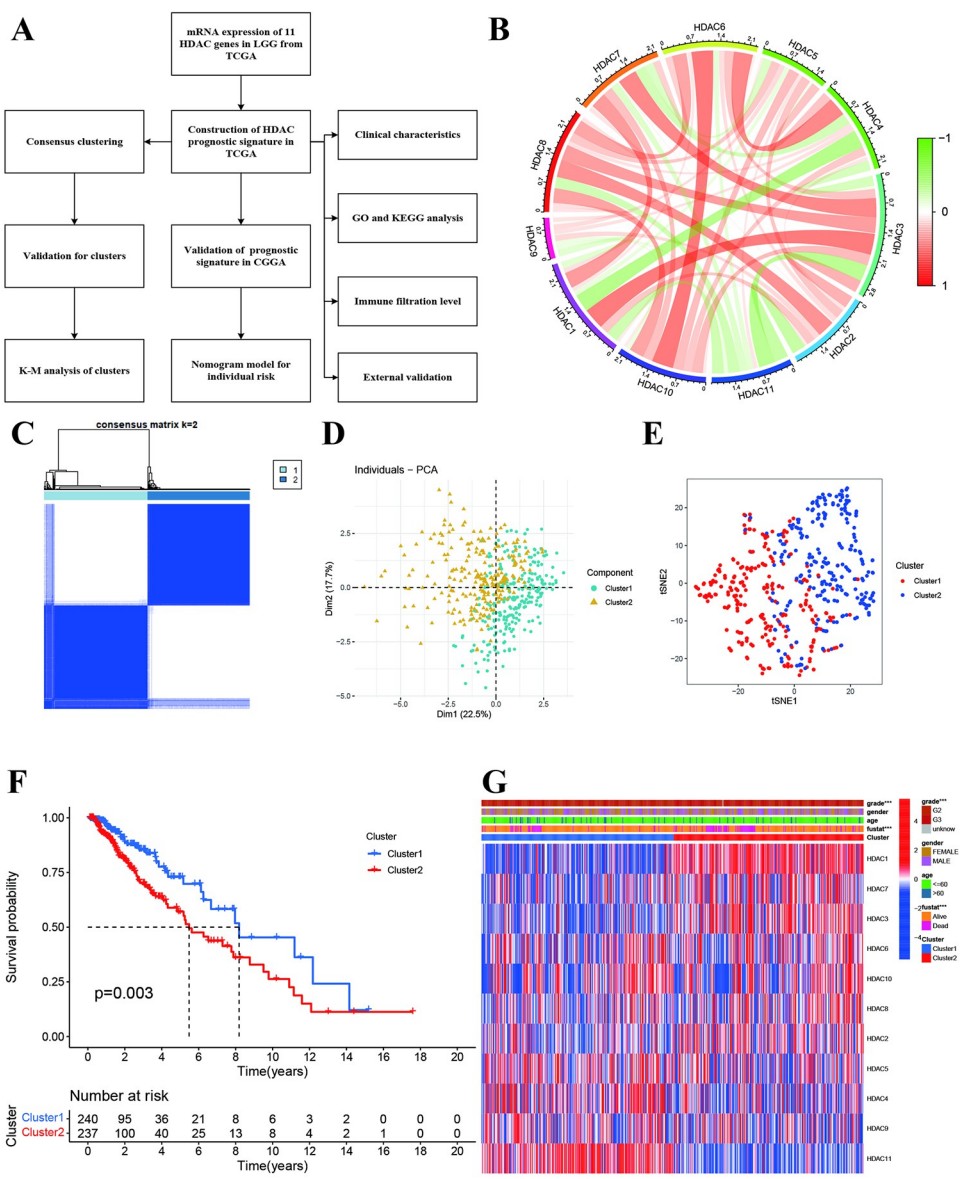

**Fig 1. Molecular subtypes for low-grade glioma based on HDAC genes. A:** The flow chart of overall data processing. **B:** Circle plot showed the correlations among HDAC genes. **C:** Consensus matrix identified two clusters. D: Principal component analysis showed two components. **E:** tSEN further confirmed two molecular clusters. **F:** Kaplan-Meier survival curves of two molecular clusters. **G:** Correlations of molecular clusters with clinical characteristics and HDAC expressions.

## Development and validation of HDAC genes prognostic model in low-grade glioma

We first performed the univariate cox regression and found that the elevated expressions of HDAC7, HDAC3, and HDAC1 were associated with poor prognosis while increased expressions of HDAC6, HDAC5, HDAC4, and HDAC11 were favorable for prognosis in low-grade glioma (Fig 2A). HDAC2, HDAC6, HDAC8, HDAC9, and HDAC10 were not associated with prognosis. Next, we performed the LASSO regression and identified 6 HDAC genes in the model: HADC1, HDAC4, HDAC5, HDAC7, HADC9, and HDAC10 (Fig 2B and 2C). We

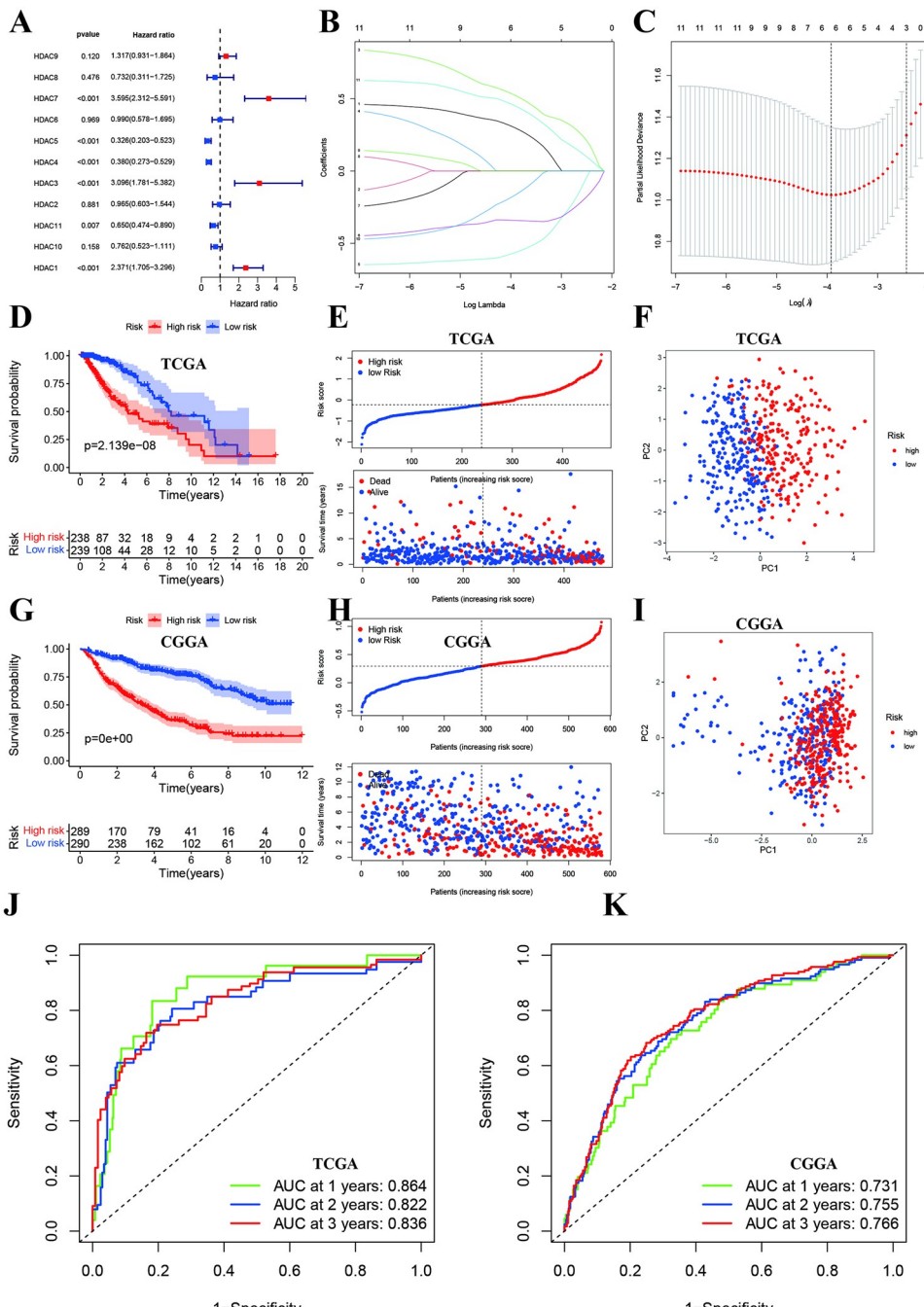

**Fig 2. Development and validations of a prognostic signature based on HDAC genes. A:** Forest plot of univariate cox regression for HDAC genes. **B and C:** LASSO regression identified the HDAC genes included in the model. **D:** Kaplan-Meier survival cures of high- and low-risk groups in TCGA. **E:** Risk score and survival time distributions of high- and low-risk groups in TCGA. **F:** PCA indicated two components in TCGA. **G:** Kaplan-Meier survival cures of high- and low-risk groups in CGGA. **H:** Risk score and survival time distributions of high- and low-risk groups in CGGA. **I:** PCA indicated two components in CGGA. **J and K:** Time-independent ROC for predicting survival at 1-year, 2-year, and 3-year using TCGA and CGGA datasets.

calculated the risk score of each sample using the following formula: risk score = 0.480 * HDAC1−0.339 * HDAC4−0.585 * HDAC5 + 0.553 * HDAC7 + 0.327 * HDAC9−0.203 * HDAC10 (S3 Table. We separated glioma patients into high-risk and low-risk group using the median of risk score. In TCGA training dataset, the high-risk group had poorer prognosis than low-risk group (Fig 2D and 2E), and PCA indicated two different components (Fig 2F). Similar trend results were found in CGGA validation dataset (Fig 2G–2I). The ROC indicated that AUCs of 1-year, 2-year and 3-year were 0.864, 0.822 and 0.836 in TCGA (Fig 2J), and the AUCs of 1-year, 2-year and 3-year were 0.731, 0.755, and 0.766 in CGGA (Fig 2K).

## Stratified analysis

Stratified analyses were performed in different clinical characteristics. Our results indicated that high-risk group still had poorer prognosis than low-risk group regardless of age (Fig 3A and 3B), gender (Fig 3C and 3D), WHO stage (Fig 3E and 3F), primary and recurrent (Fig 3G and 3H), 1p19q codeletion status (Fig 3I and 3J), and IDH mutant status (Fig 3K and 3L), chemotherapy and radiotherapy (Fig 3M–3P). These results indicated that the prognosis model was stable and was not affected by other clinical parameters.

## Clinical correlation and independent and analysis

We compared the HDACs genes expression levels between high-risk and low-risk groups, and significant differences were found in HDAC1, HDAC4, HDAC5, HDAC7, HDAC9 and

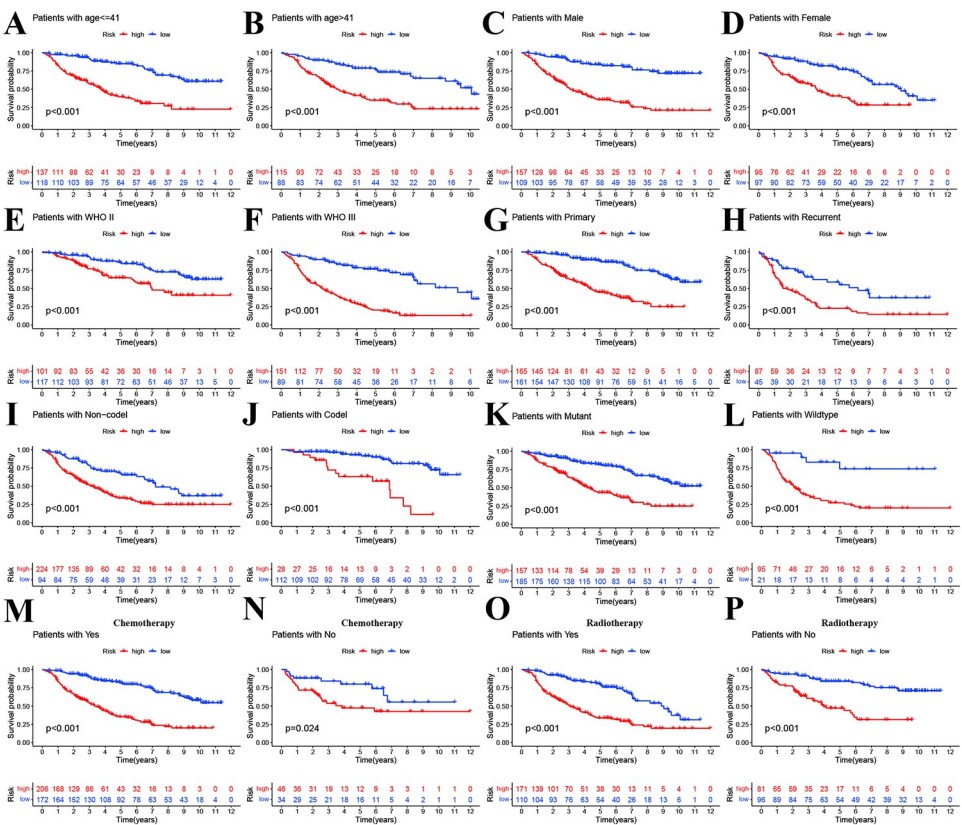

**Fig 3. Stratified analyses for prognosis of high-and low-risk groups.** A and B: age< = 41 vs age>41. C and D: Male vs Female. E and F: WHO II vs III. G and H: primary vs recurrent. I and J: 1p19q non-codeletion and codeletion. K and L: IDH mutant vs wildtype. M and N: chemotherapy Yes vs No. O and P: Radiotherapy Yes vs No.

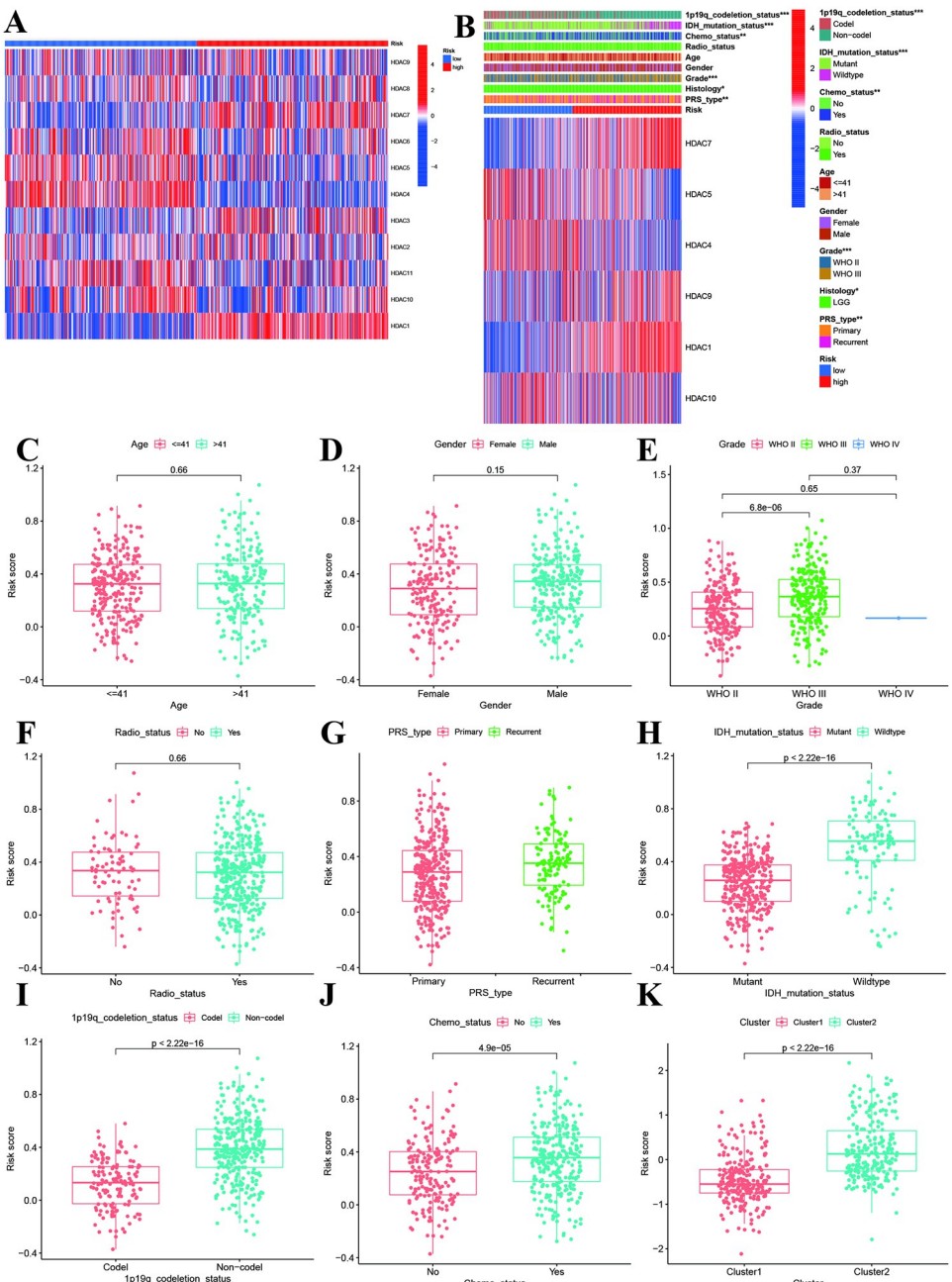

**Fig 4. Correlations of risk score with clinical characteristics. A:** Heatmap showed the expression levels of HDAC genes between high- and low-risk groups. **B:** Heatmap showed the correlations of risk groups with clinical parameters and signature genes.

HDAC10 (Fig 4A). The high-risk group tended to be advanced grade, recurrent, 1p19q non codeletion, and IDH mutant, and no history of chemotherapy (Fig 2B). No significant differences were observed in radiotherapy, age, and gender.

Then we compared the risk score differences among different clinical parameters. There were no differences in risk score of different age, gender, WHO Stage and radiotherapy (Fig 4C–4F). The results suggested that patients with advanced, recurrent, IDH wildtype, 1p19q

non-codeletion, history of chemotherapy and radiotherapy had higher risk score (P<0.05, Fig 4G–4K).

We further performed the univariate and multivariate cox regression and found risk score was an independent prognosis factor for low-grade glioma (HR:3.308, 95%CI: 2.512–4.356, P<0.001; HR:2.583, 95%CI: 1.943–3.434, P<0.001, Fig 5A and 5B), and The ROC indicated that risk score had the highest predictive ability (AUC:0.740, Fig 5C) in TCGA. Furthermore, results from CGGA also indicated that risk score was independently associated with prognosis (Univariate: HR:16.635, 95%CI: 9.721–28.466; Multi: HR:5.229, 95%CI: 2.621–10.430, Fig 5D and 5E), and AUC of risk core was 0.791 in CGGA (Fig 5F). Besides, recurrent, grade 2, chemotherapy, IDH mutant, and 1p19q codeletion were also associated with OS in low-grade glioma. We furthermore built the risk score system for in dividual patients using the nomogram (Fig 5G). We finally evaluated the fitting degrees using calibration plot, and found the predicted probability fitted with actual observed values at 1-year, 3-year and 5-year OS (Fig 5H–5J).

## Functional, pathway enrichment and mutations analysis

Using differentially expressed genes between high-risk and low-risk groups (Log fold change >1, *P*<0.05, n = 1150: 416 up-regulated and 734 down-regulated genes, S4 Table), we then performed the GO enrichment and KEGG pathway analysis to explore the functional and pathway enrichment. GO enrichment indicated that the high-risk group was mainly enriched in immunity, regulated exocytosis, cell activation in biological process, collagen-containing extracellular matrix, transport vesicle membrane, secretory granule, MHC class II protein, transport vesicle in cell component, and extracellular matrix structural constituent, gated channel activity, integrin binding, glycosaminoglycan binding, voltage-gated ion channel activity, heparin binding, immune receptor activity, and MHC class II protein complex binding in molecular function (Fig 6A). The KEGG pathways analysis indicated that the high-risk group was involved in phagosome, focal adhesion, ECM receptor interaction, cell adhesion molecules, and AGE-RAGE signaling pathway (Fig 6B). The occurrence of glioma was involved in the integrations of multiple molecular functions and signaling pathways.

We furthermore explored the gene alterations differences between high-risk and low-risk groups. The high-risk group showed high frequencies in CIC, IDH2, MUC16, SMARCA4, and FUBP1 (Fig 7A), while the gene alteration frequencies of ATRX, EGFR, PI3KA, PTEN, FLG, and FAT2 were high in low-risk group (Fig 7B). High- and low-risk groups showed similar results in variant classification, variant type, SNV class (Fig 7C and 7D). Furthermore, NIP-BL-IDH2, PIK3CA-CIC, and TP53-IDH1 showed highly co-occurrence, and PIK3CA-TP53, IDH2-TP53 showed mutually exclusive in high-risk groups. The FLG-MUC5B, OBSCN, MYOCD, MYH2, DNMT3A, COL6A3, MUC16, LRP2 showed highly co-occurrence (Fig 7E). The IDH-RP1, OBSCN, MYOCD, COL6A3, LRP2, FLG, TP53-PTEN showed mutually exclusive in low-risk group (Fig 7F).

## Tumor microenvironment, Immune filtration, and drug sensitivity analysis

We first explored the tumor microenvironment between high- and low-risk groups. we also found that the ESTIMATE score, immune score, and stromal score were higher in the high-risk group than in the low-risk group (Fig 8A–8C). Next, we analyzed the immune infiltration levels between high- and low-risk group. The results indicated that high-risk group had higher B cells, CD8+T cells, iDCs, macrophages, neutrophils, pDCs, T helper cells, Th1 cells, Th2 cells, TIL and Treg levels than low-risk group (Fig 8D). All immune-related function were elevated in high-risk group (Fig 8F). Pearson correlation analysis indicated that risk score was

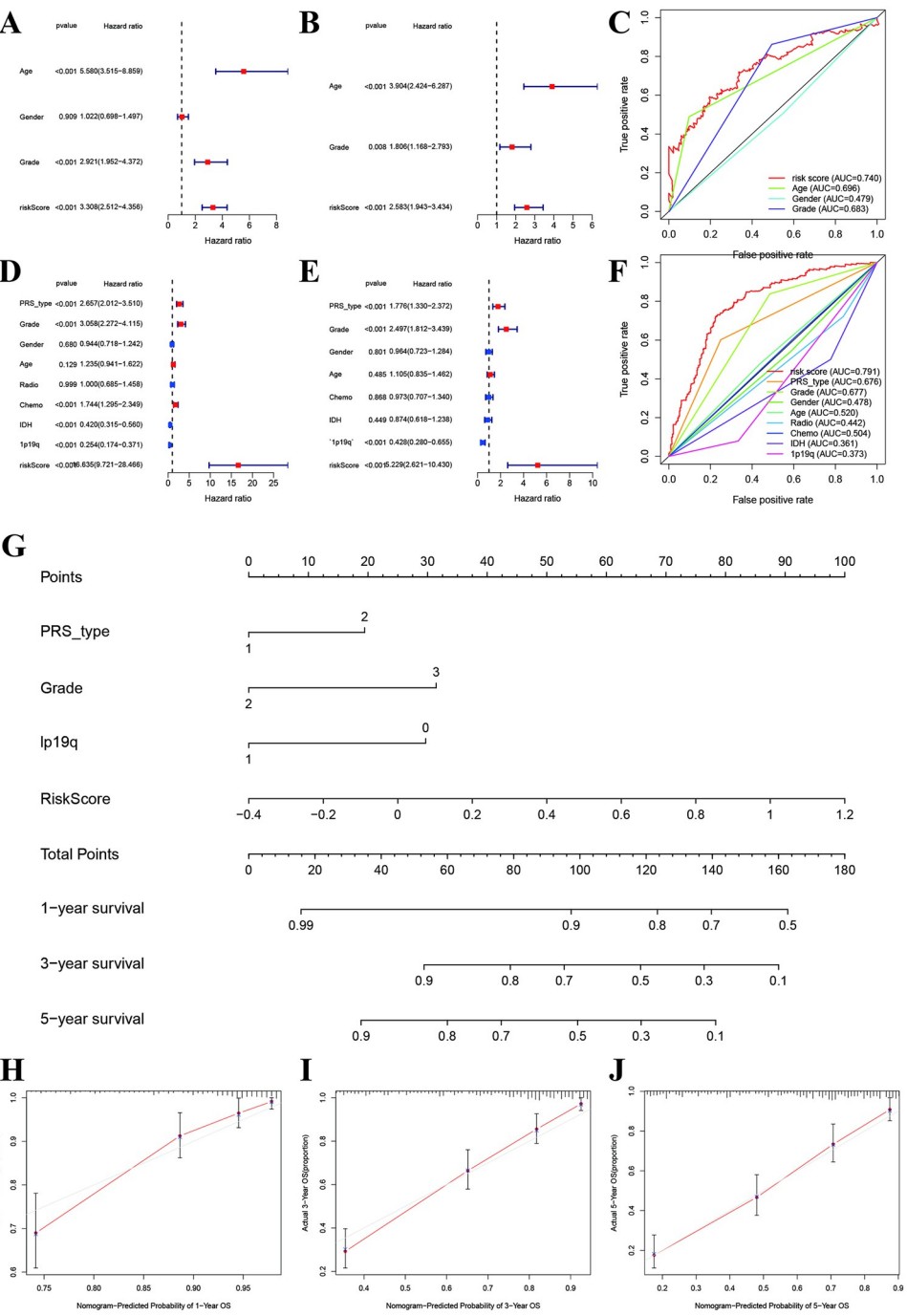

**Fig 5. Risk assessment system for induvial prognosis. A and B:** univariate and multivariate cox regression of risk score for prognosis prediction in TCGA. **C:** Multi-ROC comparisons for predicting prognosis in TCGA. **D and E:** univariate and multivariate cox regression of risk score for prognosis prediction in CGGA. **F:** Multi-ROC comparisons for predicting prognosis in CGGA. **G:** Nomogram risk assessment system for individual prognosis. **H, I and J:** Calibrations fitting plots between actual and predicted probability at 1-year, 3-year, and 5-year.

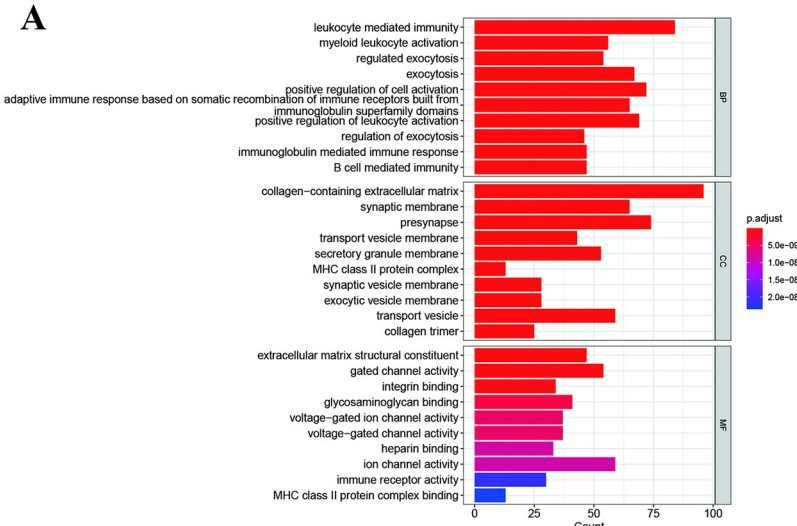

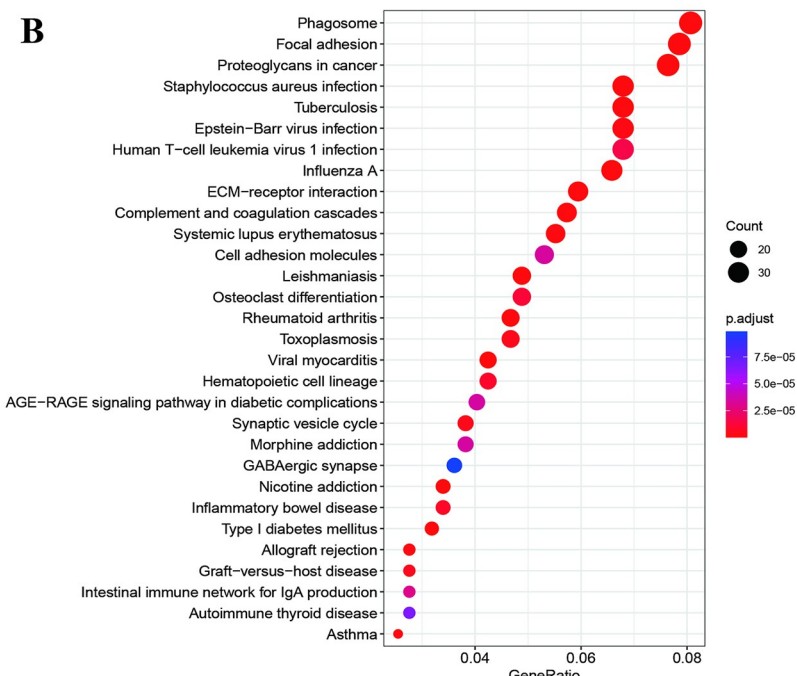

**Fig 6. Function and pathways analysis based on DGEs between high- and low-risk groups. A:** GO enrichment analysis. **B:** KEGG pathways analysis.

positively associated with Macrophages M0, M1, T cells CD4 memory activated, B cells naïve (Fig 8F–8J), while showed negatively associated with Monocytes and Mast cells activated (Fig 8J and 8K).

To explore the potential small molecular drug related with HDAC genes, we performed Pearson correlation analysis between HDAC genes expression and some small molecular drug (Fig 9). Our results indicated that HDAC7 showed significant drug resistance with Selumetinib, Cobimetinib, Evrolimus, and Rapamycin while Trametinib, PD-98059, Dabrafenib, and Dolastatin 10 showed negative associations with HDAC7. PX-316, Chelerythrine, Selumetinib

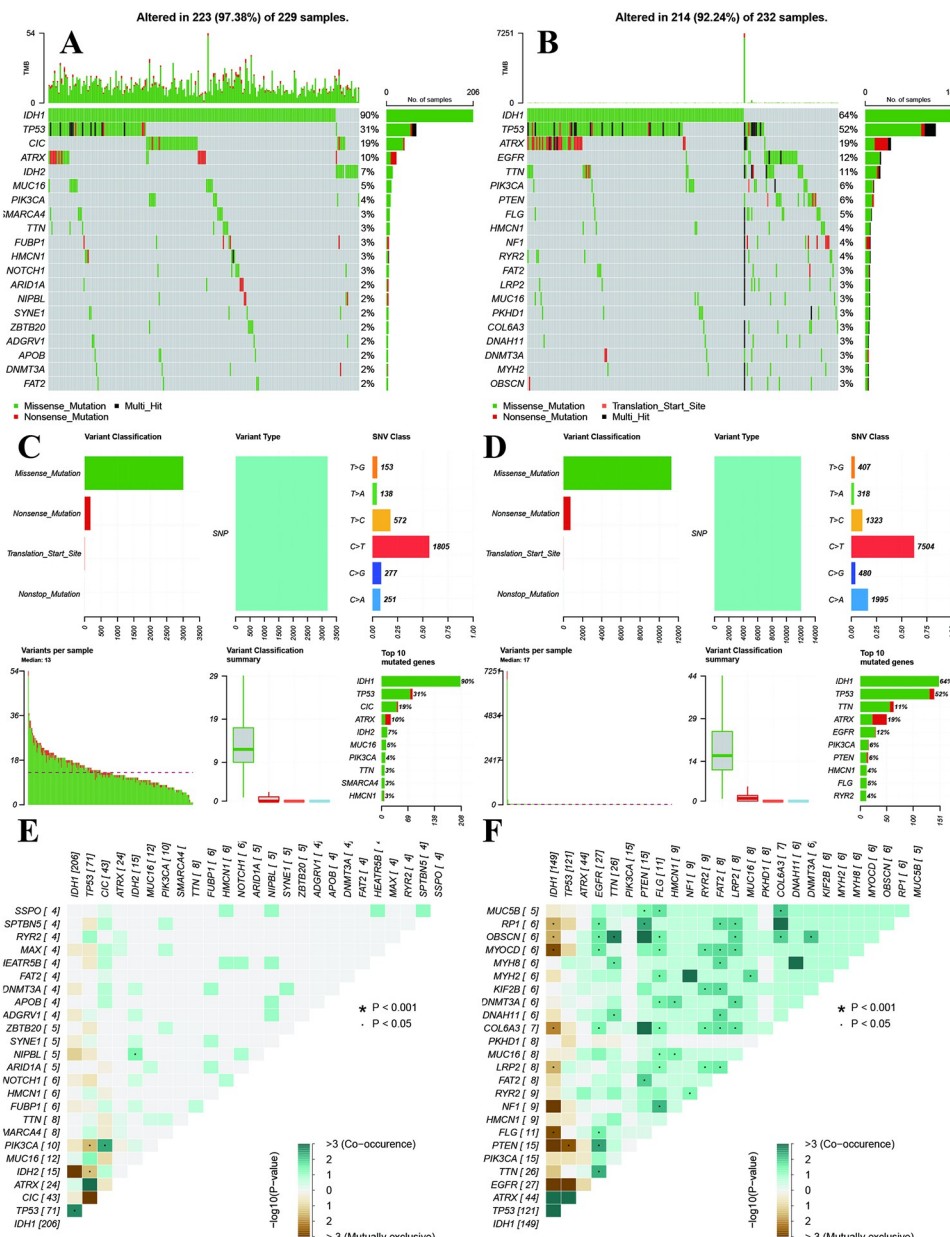

**Fig 7. Gene alterations levels between high- and low-risk groups. A and B:** Top 20 gene alterations frequencies in high- and low-risk groups. **C and D:** Variant classifications of high- and low-risk groups. **E and F:** Co-occurrence and mutually exclusive genes of high- and low-risk groups.

were positively associated with HDAC4 expression. HDAC9 showed a negative correlation with By-product o. Chelerythrine and Acrichine were also positively associated with HDAC1. These results provided some potential clues for HDAC-targeted treatment in low-grade glioma.

## Validation of HDAC genes in glioma and non-tumor tissue

Using GES420 dataset, we compared the expression levels of signature genes between tumor and normal. Our results indicated that HDAC1 was highly expressed in tumor tissues (Fig

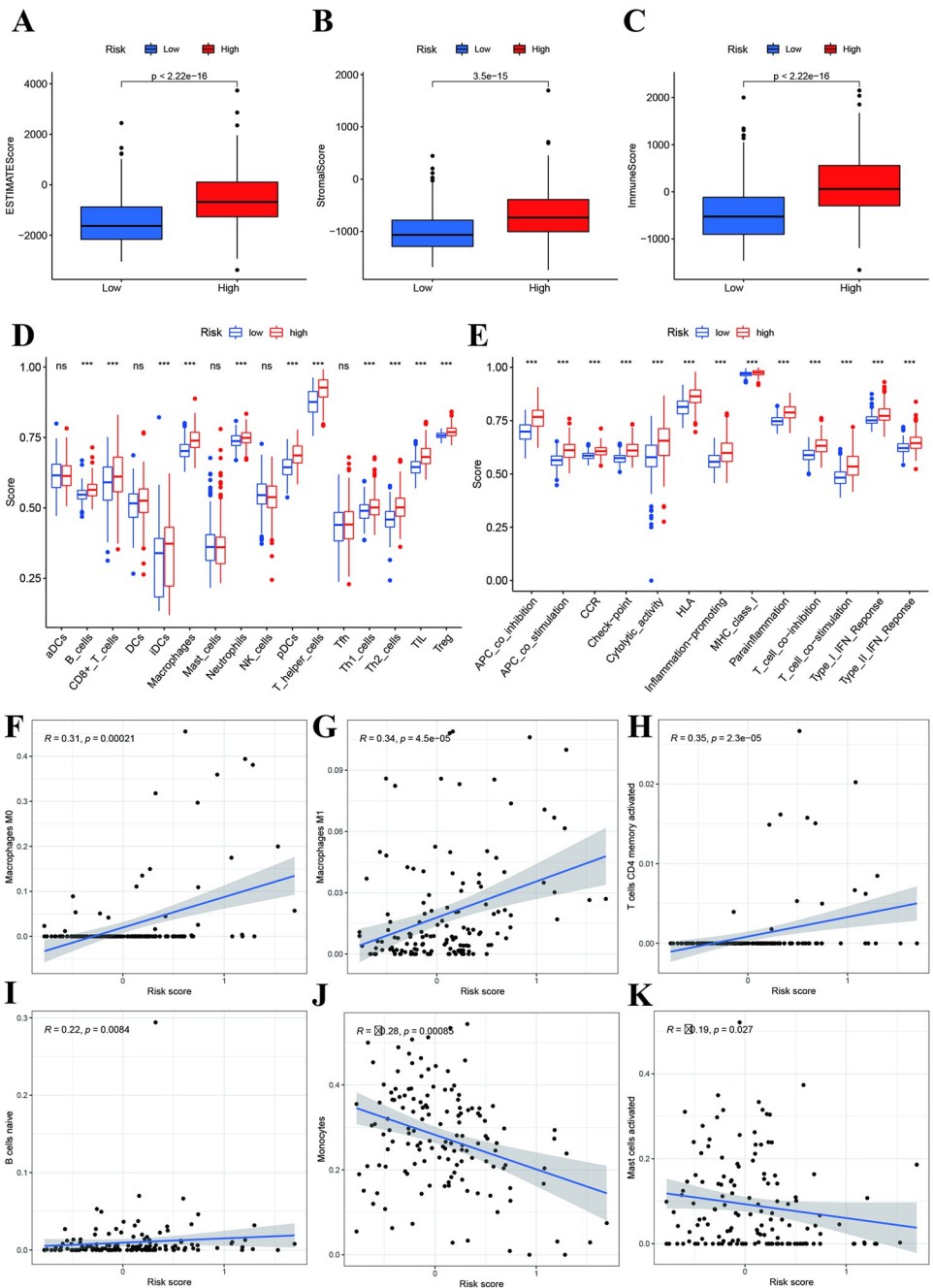

**Fig 8. Correlations of risk score with immune status. A, B and C:** Comparisons of ESTIMATE, stromal, and immune scores between two risk groups. **D:** Immune cell infiltrations between high- and low-risk groups. **E:** Immune function comparisons between high- and low-risk group. **F-K:** Scatter plot of associations between risk score and immune cells: Macrophages Mo, M1, Monocytes, Mast cells activated, T cells CD4 memory activated, and B cells naïve.

10A). There was no significant difference in HDAC4 between tumor and normal tissue (Fig 10B). HDAC5 showed decreased levels in tumor tissue (Fig 10C) while HDAC7 was significantly increased in tumor tissue (Fig 10D). HDAC9 also was lowly expressed in tumor. No significant difference was observed in HDAC10 between tumor and normal (Fig 10F).

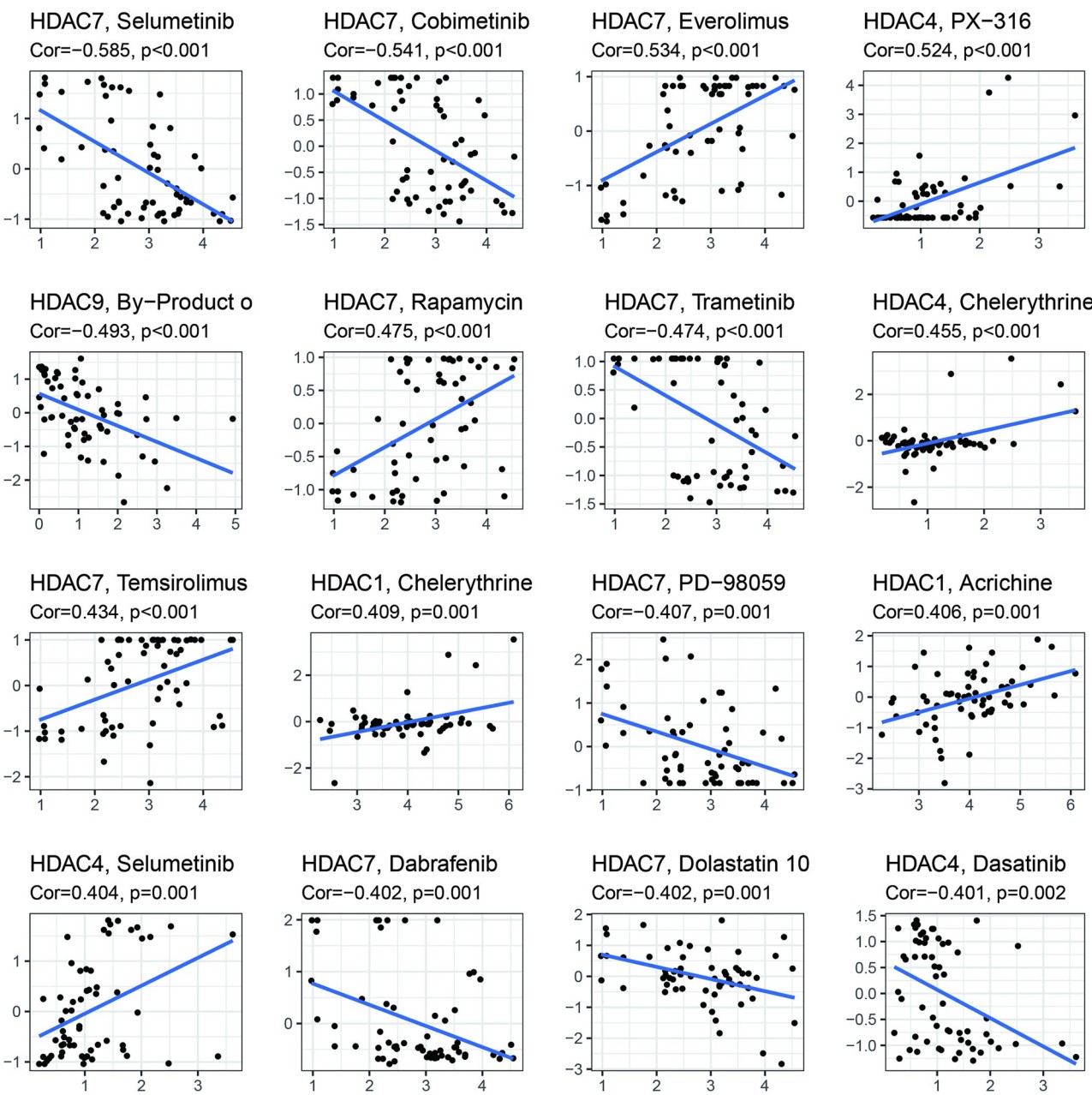

**Fig 9. Chemotherapy sensitivity analysis for signature genes: HDAC1, HDAC4, HDAC5, HDAC7, HDAC9, HDAC10.**

## Discussion

Epigenetics is the study of heritable gene expression changes that do not involve changes in DNA sequence [17]. Among its many forms, covalent modification of histones occupies an important position, which is closely related to the regulation of gene expression, including phosphorylation, acetylation, methylation modification, etc [18]. Histone acetylation and deacetylation are the most important ways and the most important driving force for gene expression regulation [19]. This reversible dynamic modification is jointly catalyzed by histone acetyltransferases (HATs) and HDACs, which together control chromatin processes. Degree

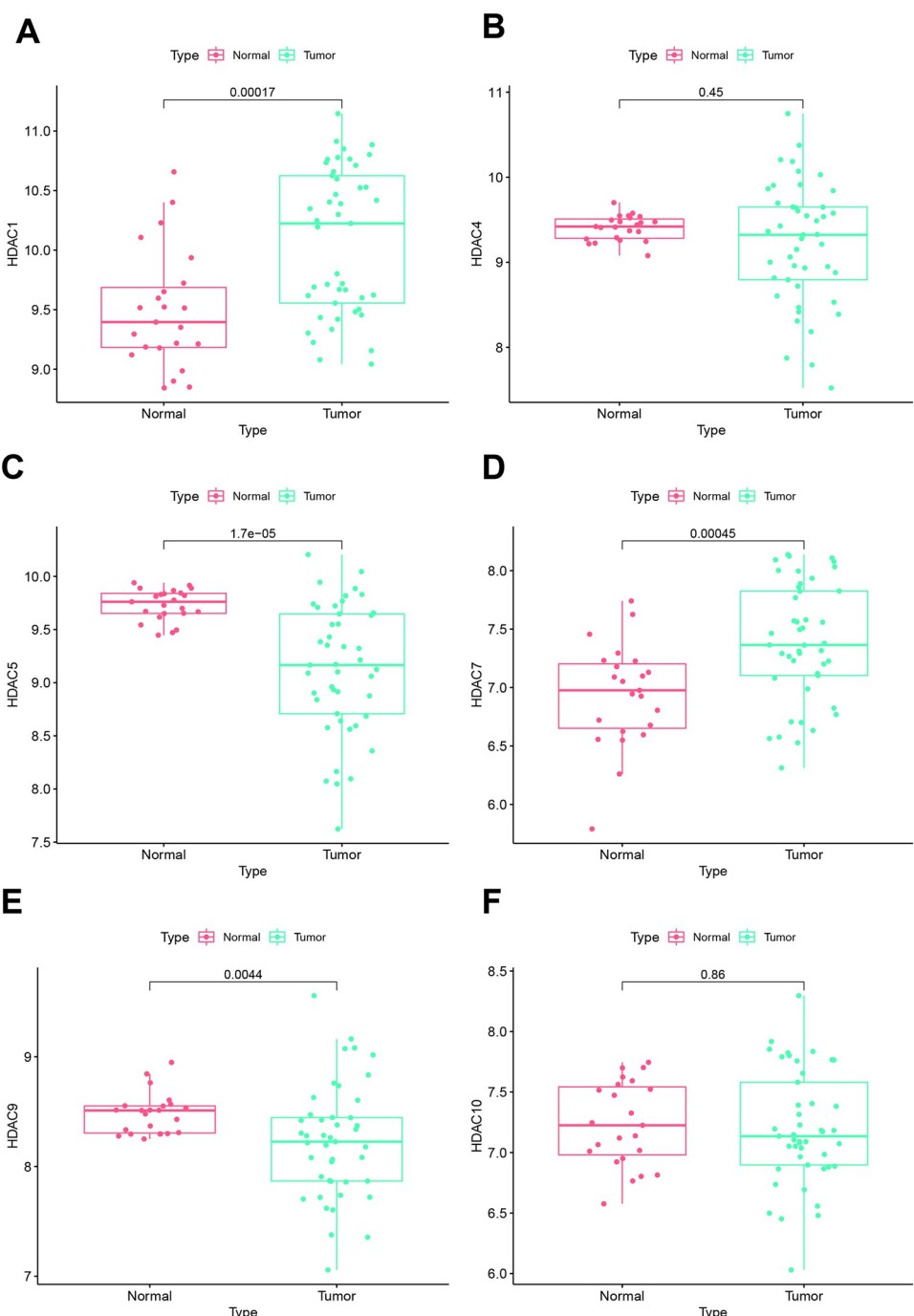

**Fig 10. Expression levels of signature genes between tumor and normal tissue. A:** HDAC1. **B:** HDAC4. **C:** HDAC5.
**D:** HDAC7. **E:** HDAC9. **F:** HDAC10.

of acetylation of regional core histones [20]. The degree of acetylation of histones is closely
related to transcriptional activity: the acetylation density of core histones in transcriptionally
active regions is high, while the acetylation density of inactive regions is low [21]. HAT pro-
motes the depolymerization of chromosomes and activates transcription; while HDACs block
DNA, thereby inhibiting the transcription process. Under normal physiological conditions,

the regulation of histone acetylation by HAT and HDACs is in a balanced state [22]. In the state of transformation, the activity of HDACs is significantly enhanced, which breaks the original balance of gene expression, resulting in the imbalance of the expression of some molecules that affect cell proliferation and cell cycle regulation, which in turn leads to malignant transformation of cells [23, 24]. HDACs have emerged as important and potential targets for antitumor drug design in epigenetics [25]. There are currently few antitumor drugs designed to act on specific targets and specific pathways. Meanwhile, it is of great of need to explore the molecular signatures for more understanding of biological relationship between tumor genotype and phenotypes.

In the present study, there were several following findings: (1) low-grade can be separated into two molecular clusters using 11 HDACs genes. Two clusters had different clinical characteristics and prognosis. Nex, we constructed a prognosis model using six HDAC genes (HDAC1, HDAC4, HDAC5, HDAC7, HDAC9, and HDAC10), which was also validated in an independent cohort dataset. Furthermore, multivariate cox regression indicated that the calculated risk score was independently associated with prognosis in low-grade glioma, and risk score can predict the five-year survival probability of low-grade glioma well. High-risk patients can be attributed to multiple complex function and molecular signaling pathways, and the genes alterations of high- and low-risk patients were significantly different. We also found that different survival outcomes of high- and low- risk patients could be involved in the differences of immune filtration level and tumor microenvironment. Subsequently, using signature genes, we identified several small molecular compounds that could be useful for low-grade glioma patients' treatment. Finally, we detected the expression levels of signature genes in tumor tissues. The present study provided a new molecular subtype way and established an overall survival prognosis model, and contributed to new insights for the understanding of molecular mechanism and treatment of low-grade glioma.

We identified six HDAC genes in established prognostic models. HDAC1 and HDAC4 belong to class I of HDAC. Previous studies have shown that HDAC1 is overexpressed in a variety of human malignancies, such as prostate cancer, breast cancer, liver cancer and lung cancer [26–28]. HDAC1 is also highly expressed in glioma tissues, and high expression of glioma is related to the proliferation, migration, invasion, angiogenesis, and poor prognosis of glioma cells [29]. In addition, it has been suggested that increased activation of HDAC1/2/6 and Sp1 is the basis of glioblastoma drug resistance and tumor growth [30]. We also found elevated HDAC1 expression in glioma tissues, which is associated with poor prognosis. HDAC4, HDAC5, HDAC7, HDAC9 and HDAC10 belong to class II of HDAC [31]. HDAC4 is often dysregulated in human malignancies, and we have demonstrated down-regulated expression in glioma tissues. However, previous studies have reported that HDAC4 is significantly upregulated in glioma tissues. Compared with U251 cells transfected with negative control, the proliferation, adenosine triphosphate (ATP) level and invasion ability of U251 cells overexpressed by HDAC4 were significantly enhanced, while U251 cells with low HDAC4 knockdown were inhibited [32, 33]. This may be related to glioma grade, stage, and histology, which needs further study. Like HDAC4, HDAC5 has also been found to be down-expressed in glioma tissues. HDAC7 acts as an oncogene in gliomas. ZNF326 has been reported to bind to specific promoter regions through its transcriptional activation domain and zinc finger structure in glioma cells to activate HDAC7 transcription [34, 35]. In addition, ZNF326 is not only highly expressed in glioma, but also positively correlated with HDAC7 expression, thus confirming the role of HDAC7 oncogene [28]. HDAC9, like most Class II HDAC, has a conserved histone deacetylase domain that catalyzes the removal of histone n-terminal tail acetyl groups and a long regulatory N-terminal domain that binds to tissue-specific transcription factors and co-repressor [36]. The amino-terminal domain contains a highly conserved serine residue. These

residues are phosphorylated. Signal-dependent phosphorylation of HDAC9 is a key event in determining cytoplasmic or nuclear localization of HDAC9. It has been reported that HDAC9 is highly expressed in many cancers [37]. In gliomas, high expression of HDAC9 promotes proliferation and tumorigenesis, in part by enhancing the EGFR signaling pathway to accelerate cell cycle [38]. Advanced tumor HDAC10 levels can be used as an indicator of resistance, and high HDAC10 levels indicate that the tumor will be resistant to chemotherapy. Compounds that specifically inhibit HDAC10 are expected to become more effective anticancer drugs to help people treat neuroblastoma, which is resistant to chemotherapy [39].

In conclusion, our study uncovers the biology function role of HDAC genes in low-grade glioma. We identified new molecular subtypes and established a prognostic model based on six HDAC genes, which was well applied in two independent cohort data. The regulation of HDAC genes in low-grade glioma involved in multiple molecular function and signaling pathways and immune infiltration levels. Further experiments in vivo and vitro were required to confirm the present findings.

## Supporting information

**S1 Table. Tripod checklist prediction model development and validation.**
(DOCX)

**S2 Table. Clustering of low-grade glioma based on 11 HDAC genes in TCGA.**
(XLSX)

**S3 Table. Coefficient of HDAC genes in the included model.**
(XLSX)

**S4 Table. Differentially expressed genes between high-and low-risk group.**
(XLSX)

## Author Contributions

**Data curation:** Lin Shen, Yanyan Li.

**Funding acquisition:** Zhanzhan Li.

**Investigation:** Na Li.

**Methodology:** Yanyan Li, Na Li.

**Project administration:** Liangfang Shen.

**Software:** Zhanzhan Li.

**Supervision:** Zhanzhan Li.

**Visualization:** Zhanzhan Li.

**Writing – original draft:** Lin Shen.

**Writing – review & editing:** Yanyan Li, Na Li, Liangfang Shen, Zhanzhan Li.

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
