## [Decision Letter · Decision Letter 0]

1 Jul 2022

PONE-D-22-04684Comprehensive analysis of Histone deacetylases genes in the prognosis and immune infiltration of glioma patientsPLOS ONE

Dear Dr. Li,

Thank you for submitting your manuscript to PLOS ONE. After careful consideration, we feel that it has merit but does not fully meet PLOS ONE’s publication criteria as it currently stands. Therefore, we invite you to submit a revised version of the manuscript that addresses the points raised below by the reviewers during the review process.

We look forward to receiving your revised manuscript.

Kind regards,

Surinder K. Batra

Academic Editor

PLOS ONE

Journal Requirements:

This study was supported by the National Natural Science Foundation of China (LZZ: 82003239), and Science Foundation of Xiangya Hospital for Young Scholar (LZZ: 2018Q012), and National Natural Science Foundation of China (SLF:81974466).

The author(s) received no specific funding for this work

Reviewers' comments:

Reviewer's Responses to Questions

**Comments to the Author**

1. Is the manuscript technically sound, and do the data support the conclusions?

Reviewer #1: Yes

Reviewer #2: Yes

2. Has the statistical analysis been performed appropriately and rigorously? 

Reviewer #1: Yes

Reviewer #2: I Don't Know

3. Have the authors made all data underlying the findings in their manuscript fully available?

Reviewer #1: Yes

Reviewer #2: Yes

4. Is the manuscript presented in an intelligible fashion and written in standard English?

Reviewer #1: Yes

Reviewer #2: No

5. Review Comments to the Author

Reviewer #1: In this study, the authors performed a Comprehensive analysis of Histone deacetylases genes in the prognosis and immune infiltration of glioma patients. Although the manuscript provides interesting findings, the way these are described need thorough editing. Below a number of points that have to be addressed before publication can be considered.

Introduction: Line no: 80 HWO should re-type as WHO.

Line no: 82 Reference 5 is not matching with the text.

Line no: 97 Reference 10 is not matching with the text. That paper talks about class I HDACs (HDAC1, HDAC2, HDAC3 and HDAC 8) not Class IIa HDACs (HDAC4, HDAC5 and HDAC7).

Figure legend Fig 1D and Fig 1E not matching with the figure. Figure legends 1F & 1G are missing.

In Fig 2A, the hazard ratio of HDAC2 and HDAC9 are almost similar. It’s not clear why they excluded HDAC2 from high risk group. (Moreover, HDAC1 and HDAC2 levels are significantly high in GBM when compared to normal brain- GEPIA data base).

Line no: 224- Typo error-The HDAC2, HDAC2 and HDAC10.

Line no:255; Should read as patients in TCGA and CGGA respectively.

Line no: 260; Should read as IDH mutation status, 1p19Q codeletion.

Line no: 269; Should read as IDH mutant and wild type.

Line no: 274 Should read as occurrence (primary or secondary).

Figure 5G-K not described in the results section.

Line no: 316: should read as year.

Line no:391: HDACs, as key enzymes that catalyze the acetylation of histones,???. Please correct the sentence.

Reviewer #2: This study by Shen et al. demonstrates that glioma patients can be divided into two subclasses

based on expression of eleven histone deacetylase (HDAC) genes, and patients from two subclasses had markedly different survival outcomes. Using multiple available data sets and bioinformatics approaches ((simple molecular subtyping and prognosis prediction-based methods), they established a prognostic model and identified six HDAC genes set (HDAC1, HDAC3, HDAC4, HDAC5, HDAC7, and HDAC9), as an independent prognostic factor in glioma patients. Furthermore, the calculated risk score which can predict well the five-year overall survival of glioma patients. It was also found that different survival outcomes of high- and low- risk patients could be linked with the differences of immune filtration level and tumor microenvironment. Finally, they also validated the prognostic model by measuring the expression levels of HDAC genes in RNA isolated form glioma patients’ tumors and non-tumor tissues samples.

This is a comprehensive analysis of glioma patients’ HDAC gene expression using TCGA and other sources data sets. This study provides new information that expression of a set of HDAC genes can be used to stratification of Glioma patients. Most of the conclusions are well supported by the results and presented data. However, there are few minor points which need further attention.

Minor comments:

Figure 9, the Pearson correlation analysis between different HDAC expression and patients’ resistance to different types of drugs is interesting. However, it is not clear while their data distributions patterns look very different, the correlation values for most of HDAC are very similar. For example, correlation of HDAC7 with Trametinib and HDAC4 with chelerythrine are very close p=0.47 and 0.45 but their distribution is very different. Please explain.

Figure 10, validation of the data by RT-PCR. Were these 23 samples of epilepsy patients used as normal brain tissues or there were from tumor adjacent tissues? It is not clear why the relative mRNA expression levels in Y axis have been presented in many arbitrary numbers such as 7, in some cases 8.5 for different HDAC. Some uniformity should be maintained in data presentation.

Page 12, line 121, HDAC2 was written two twice.

Legends of the corresponding figure has been embedded in the text of the results section. They need to be in a sperate section (Figure legend) in the text.

6. PLOS authors have the option to publish the peer review history of their article (what does this mean?). If published, this will include your full peer review and any attached files.

Reviewer #1: No

Reviewer #2: No

---

## [Author Response · Author response to Decision Letter 0]

26 Jul 2022

Dear Editor and Reviewers,

We have revised our manuscript according to the comments mentioned by Editor and Reviewers. The responses to all comments are as follows:

Responses to Journal Requirements:

Comment 1. Please ensure that your manuscript meets PLOS ONE's style requirements, including those for file naming. The PLOS ONE style templates can be found at 

https://journals.plos.org/plosone/s/file?id=wjVg/PLOSOne_formatting_sample_main_body.pdfand
https://journals.plos.org/plosone/s/file?id=ba62/PLOSOne_formatting_sample_title_authors_affiliations.pdf

Response 1: Yes, we have prepared our manuscript according to the requirements. 

Comment 2. Please note that PLOS ONE has specific guidelines on code sharing for submissions in which author-generated code underpins the findings in the manuscript. In these cases, all author-generated code must be made available without restrictions upon publication of the work. Please review our guidelines at https://journals.plos.org/plosone/s/materials-and-software-sharing#loc-sharing-code and ensure that your code is shared in a way that follows best practice and facilitates reproducibility and reuse.

Response 2: Yes, all original code can be available from the corresponding author (Zhanzhan Li: lizhanzhan@csu.edu.cn). We have stated this in Data Availability Statement.

Comment 3. We note that the grant information you provided in the ‘Funding Information’ and ‘Financial Disclosure’ sections do not match. When you resubmit, please ensure that you provide the correct grant numbers for the awards you received for your study in the ‘Funding Information’ section.

Response 3: Yes, we have revised “Funding Information”

Comment 4. Thank you for stating the following in the Acknowledgments Section of your manuscript: This study was supported by the National Natural Science Foundation of China (LZZ: 82003239), and Science Foundation of Xiangya Hospital for Young Scholar (LZZ: 2018Q012), and National Natural Science Foundation of China (SLF:81974466). We note that you have provided funding information that is not currently declared in your Funding Statement. However, funding information should not appear in the Acknowledgments section or other areas of your manuscript. We will only publish funding information present in the Funding Statement section of the online submission form. Please remove any funding-related text from the manuscript and let us know how you would like to update your Funding Statement. Currently, your Funding Statement reads as follows: The author(s) received no specific funding for this work Please include your amended statements within your cover letter; we will change the online submission form on your behalf.

Response 4: We have removed funding-related text in the manuscript. We added updated information in Cover letter. 

Comment 5. PLOS requires an ORCID iD for the corresponding author in Editorial Manager on papers submitted after December 6th, 2016. Please ensure that you have an ORCID iD and that it is validated in Editorial Manager. To do this, go to ‘Update my Information’ (in the upper left-hand corner of the main menu), and click on the Fetch/Validate link next to the ORCID field. This will take you to the ORCID site and allow you to create a new iD or authenticate a pre-existing iD in Editorial Manager. Please see the following video for instructions on linking an ORCID iD to your Editorial Manager account: https://www.youtube.com/watch?v=_xcclfuvtxQ

Response 5: We have linked ORCID ID to Editorial Manager account. 

Response to Reviewers

Response to Reviewer #1

In this study, the authors performed a Comprehensive analysis of Histone deacetylases genes in the prognosis and immune infiltration of glioma patients. Although the manuscript provides interesting findings, the way these are described need thorough editing. Below a number of points that have to be addressed before publication can be considered.

Comment 1: Introduction: Line no: 80 HWO should re-type as WHO.

Response 1: Yes, we have revised this error. 

Comment 2: Line no: 82 Reference 5 is not matching with the text. Line no: 97 Reference 10 is not matching with the text. That paper talks about class I HDACs (HDAC1, HDAC2, HDAC3 and HDAC 8) not Class IIa HDACs (HDAC4, HDAC5 and HDAC7).

Response 2: Thank you for comments. We have rewritten this paragraph and updated the references as follows: Many studies have found that abnormal expression of HDAC is related to the occurrence and development of tumors. HDAC includes four classes: class I (HDAC1, HDAC2, HDAC3, HDAC8), class II (HDAC4, HDAC5, HDAC6, HDAC7, HDAC9, HDAC10), class III (SIRT1-SIRT7), and class IV (HDAC11)[12].

12. de Ruijter AJ, van Gennip AH, Caron HN, Kemp S, van Kuilenburg AB. Histone deacetylases (HDACs): characterization of the classical HDAC family. Biochem J. 2003;370(Pt 3):737-49. http://doi.org/10.1042/BJ20021321

Comment 3: Figure legend Fig 1D and Fig 1E not matching with the figure. Figure legends 1F & 1G are missing.

Response 3: The Figure 1 had been re-prepared. Fig 1F-1G legends had been added. 

Comment 4: In Fig 2A, the hazard ratio of HDAC2 and HDAC9 are almost similar. It’s not clear why they excluded HDAC2 from high-risk group. (Moreover, HDAC1 and HDAC2 levels are significantly high in GBM when compared to normal brain- GEPIA data base).

Response 4: Thank you for your comment. Fig 2A only presented the prognosis role of single HDAC gene in glioma. The model gene identification was achieved by LASSO regression for excluding multilinearity among variables. The differences between tumor and normal tissue did not make sure that they would be also associated with prognosis. 

Comment 5: Line no: 224- Typo error-The HDAC2, HDAC2 and HDAC10. Line no:255; Should read as patients in TCGA and CGGA respectively. Line no: 260; Should read as IDH mutation status, 1p19Q codeletion. Line no: 269; Should read as IDH mutant and wild type. Line no: 274 Should read as occurrence (primary or secondary).

Response 5: Thank you for your comments. We have revised these descriptions according to your advice. 

Comment 6: Figure 5G-K not described in the results section.

Response 6: Fig5G-5J had been described as follows: We furthermore built the risk score system for in dividual patients using the nomogram (Fig 5G). We finally evaluated the fitting degrees using calibration plot, and found the predicted probability fitted with actual observed values at 1-year, 3-year and 5-year OS (Fig 5H-5J).

Comment 7: Line no: 316: should read as year. Line no:391: HDACs, as key enzymes that catalyze the acetylation of histones,???. Please correct the sentence.

Response 7: We have revised these errors. We also polished our manuscript with the help of a pro institute. The Editing Certificate have been uploaded.

Response to Reviewer #2: 

This study by Shen et al. demonstrates that glioma patients can be divided into two subclasses based on expression of eleven histone deacetylase (HDAC) genes, and patients from two subclasses had markedly different survival outcomes. Using multiple available data sets and bioinformatics approaches ((simple molecular subtyping and prognosis prediction-based methods), they established a prognostic model and identified six HDAC genes set (HDAC1, HDAC3, HDAC4, HDAC5, HDAC7, and HDAC9), as an independent prognostic factor in glioma patients. Furthermore, the calculated risk score which can predict well the five-year overall survival of glioma patients. It was also found that different survival outcomes of high- and low- risk patients could be linked with the differences of immune filtration level and tumor microenvironment. Finally, they also validated the prognostic model by measuring the expression levels of HDAC genes in RNA isolated form glioma patients’ tumors and non-tumor tissues samples. This is a comprehensive analysis of glioma patients’ HDAC gene expression using TCGA and other sources data sets. This study provides new information that expression of a set of HDAC genes can be used to stratification of Glioma patients. Most of the conclusions are well supported by the results and presented data. However, there are few minor points which need further attention.

Minor comments:

Comment 1, Figure 9, the Pearson correlation analysis between different HDAC expression and patients’ resistance to different types of drugs is interesting. However, it is not clear while their data distributions patterns look very different, the correlation values for most of HDAC are very similar. For example, correlation of HDAC7 with Trametinib and HDAC4 with chelerythrine are very close p=0.47 and 0.45 but their distribution is very different. Please explain.

Response 1: Than you for your advice. Correlation coefficient of HDAC7 with Trametinib was -0.474, and the correlation coefficient of HDAC4 with Chelerythrine was 0.455. They were negatively and positively correlated, respectively. 

Comment 2, Figure 10, validation of the data by RT-PCR. Were these 23 samples of epilepsy patients used as normal brain tissues or there were from tumor adjacent tissues? It is not clear why the relative mRNA expression levels in Y axis have been presented in many arbitrary numbers such as 7, in some cases 8.5 for different HDAC. Some uniformity should be maintained in data presentation.

Response 2: Yes, 23 samples from epilepsy patients are used as non-tumor samples. We have re-prepared the Figure 10 and the Y axis have been uniformity in data presentation.

Comment 3: Page 12, line 121, HDAC2 was written two twice.

Response 3: Yes, we have corrected this error.

Comment 4: Legends of the corresponding figure has been embedded in the text of the results section. They need to be in a sperate section (Figure legend) in the text.

Response 4: Thank you for your advice. The Journal guidelines required us to put Figure legends in the text where they were firstly cited.

---

## [Decision Letter · Decision Letter 1]

29 Sep 2022

Comprehensive analyses reveal the role of histone deacetylase genes in prognosis and immune response in low-grade glioma

PONE-D-22-04684R1

Dear Dr. Li,

We’re pleased to inform you that your manuscript has been judged scientifically suitable for publication and will be formally accepted for publication once it meets all outstanding technical requirements.

Kind regards,

Jinsong Zhang

Academic Editor

PLOS ONE

Additional Editor Comments (optional):

Reviewers' comments:

Reviewer's Responses to Questions

**Comments to the Author**

1. If the authors have adequately addressed your comments raised in a previous round of review and you feel that this manuscript is now acceptable for publication, you may indicate that here to bypass the “Comments to the Author” section, enter your conflict of interest statement in the “Confidential to Editor” section, and submit your "Accept" recommendation.

Reviewer #1: All comments have been addressed

2. Is the manuscript technically sound, and do the data support the conclusions?

Reviewer #1: Yes

3. Has the statistical analysis been performed appropriately and rigorously? 

Reviewer #1: Yes

4. Have the authors made all data underlying the findings in their manuscript fully available?

Reviewer #1: Yes

5. Is the manuscript presented in an intelligible fashion and written in standard English?

Reviewer #1: Yes

6. Review Comments to the Author

Reviewer #1: Figure legends are fine now.

Manuscript typo errors have been rectified.

All comments have been addressed. Congratulations.

7. PLOS authors have the option to publish the peer review history of their article (what does this mean?). If published, this will include your full peer review and any attached files.

Reviewer #1: No
